# Employee Participation in Workplace Vaccination Campaigns: A Systematic Review and Meta-Analysis

**DOI:** 10.3390/vaccines10111898

**Published:** 2022-11-10

**Authors:** Maria Rosaria Gualano, Paolo Emilio Santoro, Ivan Borrelli, Maria Francesca Rossi, Carlotta Amantea, Antonio Tumminello, Alessandra Daniele, Flavia Beccia, Umberto Moscato

**Affiliations:** 1School of Medicine, Saint Camillus International University of Health Sciences, UniCamillus, 00131 Rome, Italy; 2Leadership in Medicine Research Center, Università Cattolica del Sacro Cuore, 20123 Rome, Italy; 3Center for Global Health Research and Studies, Università Cattolica del Sacro Cuore, 00168 Rome, Italy; 4Department of Woman and Child Health and Public Health, Fondazione Policlinico Universitario A. Gemelli IRCCS, 00168 Rome, Italy; 5Department of Health Science and Public Health, Università Cattolica del Sacro Cuore, 00168 Rome, Italy; 6Section of Occupational Health, Department of Life Sciences and Public Health, Università Cattolica del Sacro Cuore, 00168 Rome, Italy; 7Section of Hygiene, Department of Life Sciences and Public Health, Università Cattolica del Sacro Cuore, 00168 Rome, Italy

**Keywords:** occupational medicine, vaccine acceptance, workplace vaccination campaign, vaccine hesitancy, worksite wellness, disease prevention

## Abstract

To reduce vaccine-preventable diseases in workers, workplace vaccination campaigns can be implemented on-site. The aim of this systematic review was to evaluate adherence to workplace vaccination campaigns. Three databases, PubMed, ISI Web of Science, and Scopus, were screened systematically for articles in English or Italian addressing participation in an on-site vaccination program by employees. The following data was extracted: authors, year, country, type of vaccine, type of workplace, sample size, vaccination rate. Data on the prevalence of flu vaccination were calculated and pooled using a random-effects model. Thirteen articles were included in the review, ten in the meta-analysis. Most studies were conducted in the USA (30.7%) and most vaccination campaigns were against influenza (69.2%), with a pooled estimate of 42% (95% CI: 0.25–0.60%); participation rate was highly variable, ranging from 88.9% for an influenza vaccination campaign to 5.7% for a Lyme disease vaccination campaign. Offering free on-site vaccination can be a successful tool to ensure adherence to vaccination campaigns and administration of all required doses according to the vaccine administration scheme. The occupational physician can play a key role in implementing workplace campaigns for employee mandatory vaccinations.

## 1. Introduction

In the occupational setting, exposure to biological agents constitutes a risk in various professions and can be assessed based on the environmental setting of the workplace and the specific tasks carried out by the workers. It is important for employers to protect all workers who are not immune to the biological agents that pose a risk to them to reduce the occupational biological risk in susceptible workers [1].

An effective measure to minimize or reduce biological risks is to provide workers with safe and effective vaccinations based on their specific tasks, following an appropriate assessment and evaluation performed by an occupational physician. Workplace vaccination campaigns are defined in this review as vaccination programs implemented by the employer to offer free on-site vaccination to their employees. To reduce the risk of occupational acquisition of vaccine-preventable diseases (VPDs), interventions should be tailored to the political, social, cultural, economic and occupational context [2]. Recently, the topic of workplace vaccination has been addressed in the context of the COronaVIrus Disease 19 (COVID-19) pandemic, as policies were implemented to establish severe acute respiratory syndrome coronavirus 2 (SARS-CoV-2) vaccination as mandatory, to safeguard employees’ health and address the related issue of medical liability for vaccinating healthcare professionals [3,4,5].

The prevention of VPDs is not only a necessary tool to ensure the safety of employees in the workplace, but also a cost-effective tool for employers. For example, influenza epidemics have been successfully linked to patterns of employee absenteeism. Each year, up to 10% of unvaccinated adults contract seasonal influenza and half of this percentage develop symptoms [6]. Workplace influenza vaccination campaigns are cost-saving for the company—up to EUR 314 can be saved by the employer for each employee that agrees to be vaccinated against influenza according to an Italian study (considering the costs sustained by the organize to carry out the vaccination campaign against the savings obtained by reduction in sick leave) [7]. Another study, conducted in Colombia, highlighted that between USD 89 and USD 327 were saved by the company for each vaccinated employee participating in a workplace vaccination campaign against influenza [8].

As reported in the latest WHO/UNICEFF dashboard update of 15 July 2019, the first dose of tetanus–diphtheria–pertussis (Tdap) vaccination does not appear to be routinely provided globally. Some countries, such as South America and India, have inadequate vaccination coverage, exposing workers to these biological agents on the job (e.g., farm workers, sweepers, rubbish collectors) [9].

As highlighted by Black CL et al. in 2017, vaccination coverage among workers can be improved with the active promotion of vaccination campaigns in the workplace, informing the workers of the benefits of the vaccination, and simply by offering on-site vaccination free of charge for all employees [10].

Vaccination is a key public health issue but is still controversial in the workplace, where different and somewhat complementary interests related to economic productivity, worker health protection and compliance with legislation clash and intertwine. Attention to vaccination coverage and health promotion policies and prevention campaigns are not adequately conducted and guaranteed in the workplace. As a result, absenteeism due to vaccine-preventable diseases is still a relevant problem, causing both health-related and economic issues for workers worldwide [11]. 

This problem could be addressed using two strategies: enhancing problem awareness and implementing tailored prevention campaigns. Considering the first, this systematic review, supported by meta-analysis, provides an international overview of vaccination coverage in the workplace, evaluating the adherence of workers to on-site vaccination campaigns worldwide (adherence to a vaccination campaign is defined here as the percentage of workers participating in an on-site vaccination campaign promoted by the employer). The aim of this review is to raise awareness among occupational medicine and public health professionals and to provide a starting point for policy makers by offering a perspective on strategies aimed at solving a pervasive problem for which there is a compelling scientific need.

## 2. Materials and Methods

The systematic review was performed following the Preferred Reporting Items for Systematic Reviews and Meta-Analyses (PRISMA) statement [12]. The database search was performed and updated in February 2022 and included all articles published up until 1 February 2022, with no other time limit set for the start year of publication, across three databases, PubMed, ISI Web of Knowledge, Scopus, using the following query: (“Occupational health” OR “occupational medicine” OR occupational) AND vaccin* AND (workplace OR corporate OR company OR business OR firm)

After retrieving the articles from all the selected databases, duplicate removal and initial screening by title and abstract were performed using the website tool Rayyan [13], which allowed for articles to be screened by the researchers following a triple blind methodology to reduce selection bias; any conflict between researchers was resolved by internal discussion. In order to include all pertinent articles, the word “vaccin*” was used to specify that a “vaccination campaign” had to be implemented by the employer; for the same reason the term “occupational” was used to include all terms referring to the occupational field or occupational physicians.

### 2.1. Inclusion and Exclusion Criteria

Articles written in English or Italian languages that assessed participation in free, on-site vaccination campaigns organized by the employer in the workplace were included in the review. The vaccination campaign had to be on-site and the number of employees participating in the campaign had to be stated. Articles written in another language or that did not measure the impact of workplace vaccinations or vaccination campaigns were excluded from the systematic review.

### 2.2. Data Extraction and Synthesis

The following data was extracted and reported in an Excel sheet by one of the authors (after the full text had been included by all the authors performing the screening, or internal conflict had been resolved with a decision to include the study): authors, year, country, type of vaccine, type of workplace, sample numerosity and vaccination rate. The results were presented quantitatively for: country, type of vaccination offered, sample numbers and participation rate.

### 2.3. Quality Assessment

An assessment of the methodological quality was performed on all the included studies using the Newcastle–Ottawa Scale [14].

### 2.4. Statistical Analysis

The pooled prevalence of influenza vaccination coverage in employees participating in workplace vaccination campaigns was estimated using a meta-analysis. Heterogeneity across studies was measured by the I^2^ statistic; with percentages of 25%, 50%, and 75% indicating low, moderate, and high heterogeneity, respectively [15]. A random-effects model was used if the I^2^ was larger than 50%; otherwise, a fixed-effects model was applied [16]. A forest plot was used to present the pooled prevalence. Publication bias was assessed statistically with the Egger regression model. The significance level was set at *p* < 0.05. All the statistical analyses were conducted with STATA software version 14 (Stata Corp, College Station, TX, USA).

## 3. Results

The initial search resulted in 1180 relevant articles across the three databases (i.e., PubMed, ISI Web of Knowledge, and Scopus). After removing duplicates (411 duplicate articles), the initial search resulted in 769 eligible articles. The initial screening by title and abstract resulted in a total of 722 articles being excluded based on the main topic addressed. The remaining 47 articles were screened by full text; six articles did not have a full-text version available online; therefore, 41 full-text articles were screened. After excluding 28 articles due to inadequate topic, language different than English or Italian languages, wrong publication type (e.g., editorials, commentaries, conference proceedings), or wrong outcome measure, the remaining 13 were included in our systematic review based on full text (Figure 1). Any conflict about the inclusion or exclusion of the articles was resolved by internal discussion between the researchers.

All included studies were assessed using the Newcastle–Ottawa Scale [14] and all showed at least satisfactory methodological quality (score ≥ 6).

Out of the 13 studies included in the review (Table 1), nine (69.2%) examined participation in workplace influenza vaccination campaigns [7,8,17,18,19,20,21,22,23], one (7.7%) a workplace Lyme disease vaccination campaign [24], one (7.7%) a pneumococcal vaccination campaign [25], one (7.7%) a Tdap workplace vaccination campaign [26], and one (7.7%) examined participation in two workplace vaccination campaigns for influenza and Tdap [27].

The majority of the studies, four out of 13 (30.7%), were conducted in the USA [18,19,24,27], one (7.7%) was multicentric and analyzed data globally [25]; the eight remaining studies were from different countries, reported in detail in Table 1.

Concerning sample size, three (23.0%) of the studies [8,19,27] had a sample of more than 5000 workers, four (30.8%) studies [17,20,21,23] had a sample with more than 1000, but less than 5000 workers, two (15.4%) studies [22,25] had a sample between 500 and 1000 workers, and four (30.8%) studies [7,18,24,26] had a sample of less than 500 workers. All 13 included studies were cohort studies.

Three studies about influenza [17,18,19], one about Tadp [26], one about Lyme disease [24] and one about pneumococcal [25] vaccination campaigns adopted workplace promotion beforehand, informing workers about the availability and benefits of the vaccinations offered on-site.

### 3.1. Influenza

Only one study, conducted by Elawad KH et al. [17], among healthcare workers of primary health care corporation centers in Qatar, had more than 75% of employees participating in the vaccination campaign, with 3629 out of 4082 (8.9%) being vaccinated. 

Two studies [8,22] reported between 50% and 75% of employees participating in the vaccination campaign. Santoro N et al. [8] conducted a study in a chemical company in Argentina over a period of nine years, with 3436 out of 6143 (55.9%) employees being vaccinated (the number refers to the cumulative number of vaccines and employees of the company in the nine years the study was ongoing); Morales A et al. [22] evaluated a vaccination campaign aimed at bank workers in Colombia and reported 424 out of 759 employees (55.9%) vaccinated against influenza.

Four studies [18,20,21,27] highlighted adherence to workplace vaccination campaigns ranging between 25% and 50%: Samad AH et al. [20] evaluated the cost-effectiveness of a vaccination campaign in a petrochemical company in Malaysia, with 504 out of 1022 (49.3%) employees being vaccinated against influenza; Liu YH et al. [21] evaluated participation in an electronics manufacturing company, where 925 out of 2384 (38.8%) employees took part in the vaccination campaign; and Montejo L et al. [18] conducted a study on USA retail workers, reporting 91 out of 246 (37.0%) employees participating in the campaign. In a further investigation conducted by Ostovari M et al. [27], 2835 out of 8332 (34.0%) public university employees participated in the study.

Three studies [7,19,23] highlighted adherence to the vaccination campaign of below 25% of employees. Leighton L et al. [23] performed a study in the UK, evaluating the adherence to a vaccination campaign in a service company, with 601 out of 2557 (23.5%) employees taking part in the campaign. Ofstead CL et al. [19] conducted a study in three industrial factories in the USA in which 3144 out of 13,520 (23.2%) employees participated. It is important to note that this study offered an incentive to undergo vaccination (25 dollars into the employee’s health savings account). Ferro A et al. [7] highlighted the adherence of 60 out of 408 (14.7%) employees in an Italian manufacturing company.

Ten studies were included in the meta-analysis or influenza vaccination coverage in workplace vaccination campaigns. The total number of participants in the study was 39,453. All studies scored average and above according to the Newcastle–Ottawa Scale quality assessment tool. The pooled influenza vaccination coverage was 42% (95% confidence interval (CI): 25–60%) (Figure 2). There was high heterogeneity in these studies (I^2^ = 99.93%, *p* < 0.001), without significant publication bias (Egger test = 2.05 95% CI = −55.72 to 59.83, *p* = 0.93). Considering the high heterogeneity, a random effects model was adopted. By subgroup analysis in relation to the country of reference, in the Americas (USA, Colombia and Argentina), the pooled influenza vaccination coverage for the five eligible studies was 41% (95% CI: 27–55%; I^2^ = 99.82%, *p* < 0.001).

### 3.2. Other Vaccines

Two studies evaluated the participation of workers in Tdap vaccination campaigns [26,27]: Randi BA et al. [26] performed a study among 443 hospital workers in Brazil, with 175 (39.5%) participating in the campaign, while Ostovari M et al. [27] reported 475 (5.7%) public university workers receiving the Tdap vaccination.

One study, by Donoghue AN et al. [25], assessed participation in a pneumococcal vaccination campaign in welders of a multi-national company; therefore, the study was multicentric, reporting 241 (31.4%) out of 767 employees participating in the vaccination campaign.

A study by Nolan K et al. [24] evaluated participation in a campaign offering Lyme disease vaccine to employees from different workplaces that had a professional risk of contracting Lyme disease during their professional duties (i.e., environmental health inspectors and field staff, rabies-control field research staff and rabies laboratory staff who routinely handled animals); a total of 30 (13.8%) out of 190 employees decided to be vaccinated.

## 4. Discussion

The aim of this systematic review and meta-analysis was to assess how many workers participated in workplace vaccination campaigns that were free and performed on-site. Overall, ten studies investigated influenza vaccination campaigns; among these, one study [17] reported an excellent participation rate of 88.9%, two studies [8,22] reported that 55.9% of workers decided to participate in the vaccination campaigns, four studies [18,20,21,27] included in the review reported between 25% and 50% of workers participating, and three studies [7,19,23] reported an adherence lower than 25% of employees. Two of the included studies considered Tdap workplace vaccination campaigns, with 39.5% [26] and 5.7% [27] of workers being vaccinated. One study [25] examined a pneumococcal vaccination campaign, with 31.4% of workers being vaccinated. Another study [24] considered a Lyme disease vaccination campaign, with 15.8% of workers accepting the vaccination.

From an occupational health perspective, it is interesting to note how the highest participation in a workplace vaccination campaign against influenza was achieved in a healthcare setting, considering that healthcare workers are amongst the occupations most frequently exposed to biological risks. The heterogeneity highlighted by the results of the meta-analysis may reflect the different ways in which the vaccination campaigns were conducted in the workplace and the awareness of certain occupational groups of the risk of disease. These aspects would modulate vaccination adherence and vaccine hesitancy. Furthermore, the meta-analysis performed highlighted a 42% pooled prevalence concerning participation in workplace vaccination campaigns against influenza—this showed a higher participation rate in workers than that of the general population. A systematic review with meta-analysis was performed in 2018 by Wang et al. The authors performed a pooled prevalence analysis on studies addressing vaccination coverage worldwide, highlighting a 23.2% vaccination coverage against influenza in the general population (not limited to the working population) [28]. Based on COVID-19 vaccination campaign experiences, occupational physicians can provide support to vaccination campaigns in the workplace, reducing the additional costs of vaccination in non-healthcare settings. In defining a specific and coordinated strategy to identify the setting, the procedures for vaccination management, the vaccine doses and prioritization of the most fragile workers, occupational physicians can enhance the prior knowledge of workers, overcome hesitancy and properly recognize health needs [29]. The promotion of influenza vaccination in the workplace, as well as the availability of vaccinating physicians in the occupational setting to inform workers, as well as to offer on-site vaccination (which is both cost- and time-saving for the employee) could represent essential public health tools to improve vaccination coverage.

Black CL et al. [10] reported that offering free on-site vaccination can be a successful tool to ensure adherence to vaccination campaigns. Therefore, on-site vaccination campaigns against influenza could be highly beneficial in most, if not all, workplaces. On the other hand, vaccination campaigns should be appropriately presented to the workers, with information about benefits and availability provided to employees before the start of the campaign, as was done by Elawad KH et al. [17] through promotional posters in the workplace.

Furthermore, in addition to encouraging participation in single-dose vaccination campaigns, Yuan et al. [30] highlighted that workers participating in on-site vaccinations are more likely to complete all doses required by the specific vaccine than people receiving the first dose of a multi-dose vaccination outside of their workplace. In 2019, Cassimos et al. [31] updated a comprehensive list of mandatory vaccinations for healthcare workers in Europe, pointing out the necessity of specific workplace vaccination campaigns aimed at all the mandatory vaccination types: including influenza, hepatitis B, Tdap and poliomyelitis, to name just a few. The necessity of conducting such campaigns applies not only to healthcare professionals, but to all professionals. If vaccinations are mandatory to safely perform a job task, then the vaccination should be freely provided on-site; therefore, occupational physicians should play a key role in implementing workplace vaccination campaigns.

As highlighted by the Italian National Institute for Insurance against Accidents at Work (Istituto Nazionale per l’Assicurazione contro gli Infortuni sul Lavoro, INAIL), in 2019, 50.4% of interviewed occupational physicians indicated that influenza vaccination should be mandatory for workers, since influenza represents a public health problem [32]. Therefore, it is reasonable to assume that most Italian occupational health professionals would be willing to implement on-site vaccination campaigns for workers in their care, not only for influenza, but also for vaccinations that are already mandatory for children in Italy (e.g., HBV, Tdap, MMRV), or vaccinations to prevent diseases for which the workers are at risk due to their specific work tasks. An Italian study [33] pointed out how attitudes towards compulsory vaccination are mostly dependent on information sources and confidence towards health professionals From an occupational and public health standpoint, since a relationship based on trust between the occupational physician and the workers should always be present, workplace vaccination campaigns could be an essential tool in ensuring the health and safety of all workers.

It is important to emphasize that workplace vaccination campaigns do not just represent an important prevention tool for individual workers and for the company the campaigns take place in, but that they are also effective in supporting public health strategies in general, as the working population represents a large percentage of the overall country population. Therefore, integration of workplace healthcare with the national healthcare system would be much more effective in providing everyone access to vaccination in a timely manner. 

Furthermore, workplace vaccination campaigns could become an important tool for mitigating social inequalities by offering free on-site vaccination to all workers as the cost of vaccination campaigns performed in the workplace is usually covered by the company or employer. In countries where the national healthcare system does not cover universal healthcare, implementing workplace vaccination campaigns could help to reduce social disparities, both directly (i.e., covering the cost of vaccination) and indirectly (i.e., preventing diseases in workers by offering them access to vaccination could reduce future costs they would incur if they were to get the infection).

The findings of this systematic review could represent a starting point for policy makers and stakeholders to develop and implement new workplace prevention strategies for vaccination in the workplace. Two main areas of concern with respect to implementation of new policies emerge from this review, both at a national and at a company level. First, the range of vaccinations offered in the workplace should be broadened to all VPDs that workers are exposed to during their work tasks. Second, adherence to vaccination campaigns should be implemented through active promotion by informing employees of the benefits of vaccination and by offering vaccines to all employees free of charge. This seems advisable considering, in particular, that the study included in this review with the highest participation rate in a workplace vaccination campaign implemented an information and promotion campaign [17]. Furthermore, offering free on-site vaccination can be a successful tool for encouraging adherence to vaccination campaigns and administration of all required doses according to the vaccine administration scheme [30].

In the COVID-19 pandemic context, workplace vaccination campaigns could have been a very useful public health tool. As emerged in this systematic review, no COVID-19 workplace vaccination campaign was reported in the scientific literature. This could have been because no scientific study was conducted while implementing workplace vaccination campaign against COVID-19, but it could also suggest that workplace vaccination campaigns against COVID-19 were rare. However, the importance that workplace vaccination could have had during the critical phase of the pandemic in accelerating vaccination rates should be emphasized. In addition, workplace vaccination could represent a useful tool to protect at-risk and frail (e.g., older, immunocompromised, suffering from chronic illnesses) workers.

This systematic review has certain limitations. Only articles written in the English or Italian languages were included in the review, with eight articles excluded due to the language criterion. Furthermore, the full text for several articles could not be retrieved, despite the articles initially being included based on the title and abstract, as the full-text copy was not available online. Thus, six articles were not retrieved. A further limitation is that adherence to the vaccination campaign was evaluated only for workers who were vaccinated through a workplace vaccination campaign, without accounting for employees who received the same vaccination outside of the campaign. With respect to the meta-analysis, the absence of common variables among the included studies did not allow for stratification or subgroup analysis and the source of the high heterogeneity observed could not be investigated. 

To expand on this, due to the lack of studies investigating participation in workplace vaccination campaigns for other VPDs, we were only able to perform a pooled prevalence analysis on influenza vaccination coverage. This review highlights the need to gather further data on workplace vaccination campaigns concerning different types of vaccination provided in different workplace settings, to more thoroughly determine barriers and facilitators in the uptake of vaccination (for example, higher coverage in academic contexts or in the health sector could argue in favor of education as a promoter, while lower coverage in third sector companies could be indicative of a lesser focus on worker well-being). It would be important for future studies to explore the characteristics of each type of workplace vaccination campaign in relation to the country and workplace setting to support the development and implementation of specific public health and occupational health prevention measures and strategies.

## 5. Conclusions

As emerged in this systematic review, workplace vaccination campaigns are mostly centered on influenza vaccination campaigns and participation by employee is extremely variable depending on vaccine, country and field of work. As VPDs still represent a threat to workers to this day, and as vaccinations becomes more and more available worldwide, the role of the occupational physician and of workplace vaccination campaigns should be re-imagined.

The implementation of workplace vaccination campaigns, as well as putting in place essential measures to encourage workers to participate, should result in a reduction in the risk that exposed workers are subjected to on a daily basis. Two main strategies for implementation emerge from this review: first, that the range of vaccinations offered in the workplace should be broadened to all VPDs that the workers are exposed during their work-related activities; second, that adherence to vaccination campaigns should be implemented through active promotion, by informing employees of the benefits of vaccination, and by offering vaccines to all employees free of charge.

Occupational physicians should take on a central role in the implementation of vaccination campaigns by properly informing workers about available vaccinations and the benefits of being vaccinated, and by monitoring the vaccination status of employees during medical surveillance. The occupational physician should play a central role, as they are in a unique position to conduct medical evaluations and to build doctor-patient relationships based on trust with healthy adults. Workplace vaccination campaigns represent a great opportunity to prevent VPDs in adults that may not have been properly immunized during childhood.

## Figures and Tables

**Figure 1 vaccines-10-01898-f001:**
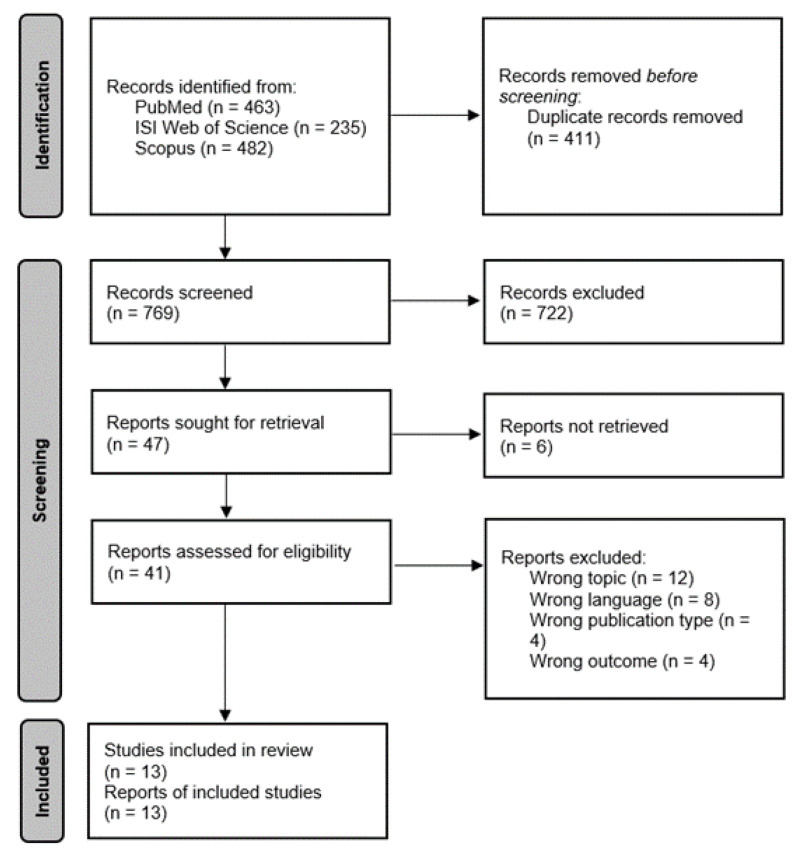
PRISMA flowchart.

**Figure 2 vaccines-10-01898-f002:**
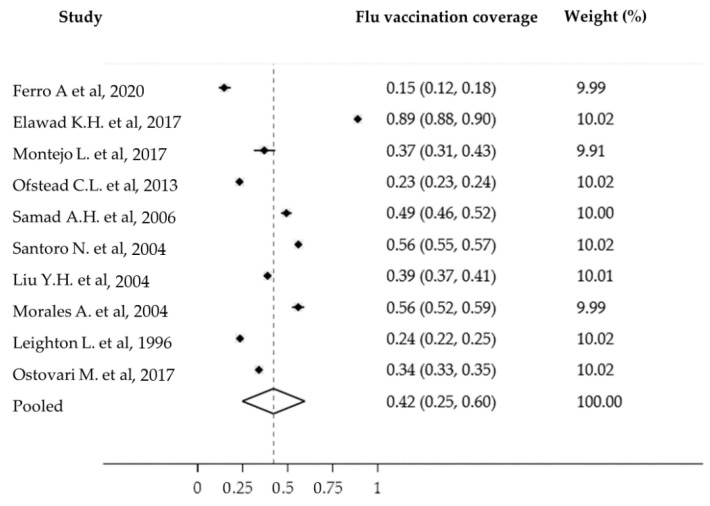
Forest plot of the eligible studies [7,8,17,18,19,20,21,22,23,24,25,26,27].

**Table 1 vaccines-10-01898-t001:** Main characteristics of the articles included in the systematic review, ordered by type of vaccination campaign and year (from most to least recent).

Author	Year	Country	Workplace	Vaccine	Participants, n	Vaccinated, n (%)
Ferro A et al. [7]	2020	Italia	Manufacturing company	Influenza	408	60 (14.7%)
Elawad KH et al. [17]	2017	Qatar	Primary health care corporation centers	Influenza	4082	3629 (88.9%)
Montejo L et al. [18]	2017	USA	Retail workers	Influenza	246	91 (37.0%)
Ofstead CL et al. [19]	2013	USA	Industrial factories	Influenza	13,520	3144 (23.2%)
Samad AH et al. [20]	2006	Malaysia	Petrochemical company	Influenza	1022	504 (49.3%)
Santoro N et al. [8]	2004	Argentina	Chemical company	Influenza	6143	3436 (55.9%)
Liu YH et al. [21]	2004	Taiwan	Electronics manufacturing company	Influenza	2384	925 (38.8%)
Morales A et al. [22]	2004	Colombia	Bank employees	Influenza	759	424 (55.9%)
Leighton L et al. [23]	1996	UK	Service company	Influenza	2557	601 (23.5%)
Nolan K et al. [24]	2006	USA	Multiple workplaces	Lyme disease	190	30 (15.8%)
Donoghue AN et al. [25]	2019	Multicentric	Welders (Multi-national company)	Pneumococcal	767	241 (31.4%)
Randi BA et al. [26]	2019	Brazil	Hospital	Tdap	443	175 (39.5%)
Ostovari M et al. [27]	2017	USA	Public university	Influenza	8332	2835 (34.0%)
				Tdap	8332	475 (5.7%)

## Data Availability

Data are available upon reasonable request.

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
