# Peer review of "Employee Participation in Workplace Vaccination Campaigns: A Systematic Review and Meta-Analysis"

_vaccines, 2022, doi:10.3390/vaccines10111898_

Round 1
Reviewer 1 Report
Major points:
Introduction:
- - It is unclear what vaccination campaign in the workplace means (both in Introduction and throughout the paper). Does it refer to providing on-site vaccination at workplace? Or does it include promoting vaccination to workers/employees? Please clarify.
Methods
- - Search terms: while the study focuses on vaccination “campaign”, the search term does not include it. Also, it is little concerning that the term “occupational’ needed to be included in addition to “workplace”, etc. Could the authors provide more rationale/justification in the methods section?
- - Please provide types of studies included in the study (e.g., cohort study, cross-sectional studies). Also, what did the authors do with grey literature or conference proceedings?
- - Was there no limit for the start year of included studies? Please provide any rationale.
- - Need to describe more in-depth regarding who did the data extraction. Was the review and data extraction done by one person or two persons? If there is any discrepancy between reviewers, how was that resolved?
- - I also suggest to include exact search terms for each database in an appendix.
Results
- - Line 156: “the adherence to a vaccination campaign” refers to getting vaccinated when vaccinations were offered on-site at the company? What about those who got vaccinated elsewhere (i.e. other than at the company)?
- - There was no study regarding COVID-19 vaccines? Please provide more details/rationale.
- - I wonder whether the percent vaccinated can be pulled together as a meta-analysis at least for influenza vaccine.
- - Table 1: Could authors add a column and include more details of the studies in the table? Such as whether vaccinations were provided at the workplace? That will really help to understanding the mechanisms and “why” some studies are reporting higher percentages of vaccinations.
Conclusions
- - Line 243-249 are more for discussion rather than conclusions of the study. Also,
Minor points
Line 43: please remove “,” after “… the biological risk”
Line 49: Change “Sars-CoV-2” to “SARS-CoV-2’
Line 52: please remove “,” after “…VPDs is”
Line 53-58: These can be better written, I believe. For example, it can start directly with the sentence “Workplace Influenza vaccination campaigns …. For the company.” Then followed by “In Italy, a company can save up to 314 euro per each employee….”
Line 61: Countries -> countries
Line 62: “leaving exposed those workers” -> “exposing the workers”
Line 63: “these biological” -> “the biological”
Line 64: Please rewrite this sentence.
Author Response
Dear Reviewer,
Thank you for accepting to review our manuscript. Regarding your valuable comments to improve our manuscript:
- Regarding the introduction, a definition for “workplace vaccination campaign” has been added in this section (lines 45-47) and has been modified in the methods section to be more precise (lines 87-88).
- Concerning the methods section:
- The term campaign was not included in the search because it would have restricted the search, to be as inclusive as possible only the term “vaccin*” was used to describe the intervention. The term “occupational” was added for the same reason; the other terms indicating the occupational field that were used in the search (“occupational health” and “occupational medicine”) did not include all terms associated with occupational vaccination campaign performed on-site (i.e.: “occupational physician” or “occupational doctor” or “occupational sector”, and so on). We have explained our rationale in the methods section (lines 86-89).
- We added the type of included studies (line 134) and stated that articles like conference proceedings or grey literature were excluded (lines 108-110).
- No limit was set for the start year of included studies, in order not to lose articles relevant to the purpose of the study, since vaccination was introduced in workplaces at different times depending on the geographical area and the vaccine considered. Any filter applied, therefore, would not have had a scientific rationale and we believe that any reduction in the articles included would not have benefited the results of the study.
- Methodology concerning conflict resolution was added (lines 85-86); a more in-depth description regarding data extraction has been added (lines 99-101).
- The search terms included in the manuscript (lines 79-81) have been already included in the manuscript, no additional filters (for example for language or publication year) were used for any database searched.
- Concerning the results section:
- An explanation of the intended meaning of “adherence to vaccination campaign” in our paper has been added in the introduction (lines 73-75). Furthermore, we added to the review’s limits the impossibility to evaluate the participation in vaccination campaigns outside the workplace campaign included in the original study (lines 242-245).
- No study were found concerning COVID-19 vaccination campaigns performed on-site and published up to February 2022 (when the initial search was performed).
- Thank you for your valuable suggestion, we have added a meta-analysis (pooled prevalence estimate) for the data regarding Influenza vaccination (Figure 1, and lines 27-28, 109-116, 183-187, 222- 226, 262-264)
- Concerning table 1, all vaccination were performed in the workplace, and we excluded studies if the vaccination was not performed on-site, as stated in the methodology; if we were to add a column in Table 1 to specify manuscript in which the vaccination were not performed in the workplace it would remain empty.
- Concerning the conclusions, we agree that the initial paragraph was a repetition of the discussion section and it has been deleted, thank you for the suggestion.
- All the minor corrections have been implemented in the manuscript, thank you.
Reviewer 2 Report
Not acceptable for publishing
Author Response
Dear reviewer,
we are sorry to read yor decision; however, according to your valuable suggestions we have modified our manuscript. We'd appreciate if you could reconsider your decision or let us know your reasons for future reference. Thank you very much.
1. The systematic review does not even register in the database
Response: Before starting our systematic review, we searched for similar work in the most widely used scientific databases and in the PROSPERO database. Vaccination in the workplace is a topic of great interest in occupational and public health medicine, but it has unfortunately not been adequately addressed by scientific research. In view of our working group's need for a clear and comprehensive view of the phenomenon, we conducted a systematic review, which was later expanded with meta-analysis at the suggestion of another reviewer. We chose not to register the systematic review on PROSPERO considering that we carried out the study in record time. registration on PROSPERO is not a required step in conducting systematic reviews, but is a choice made by some authors, often linked to the topic addressed and the timing of the study. In our opinion, the lack of registration in this case (but this would be valid for all systematic reviews) does not constitute a demerit for the study, as it is a procedural aspect not required and unrelated to the content of the study itself.
2. the contents for PRISMA flow chart for selection does not even followed.
Response: We downloaded the latest template available on the PRISMA statement website and filled it out following the instructions provided. Either way, we have redone the flowchart (Figure 1), hoping it meets your requirements.
3. lack in scientific need and originality
Response: As occupational health and public health physicians, we feel that this paper offers a first attempt to close a gap in the scientific literature related to vaccination in the occupational and public health. Vaccination is a key public health issue, but one that is still controversial in the workplace, where different and complementary drives of denialism, protection of production interests, protection of workers' health, and compliance with current legislation clash. Just as an example, in Italy, work and health are both constitutional rights, and yet we believe that attention to vaccination coverage and health promotion policies and prevention campaigns are not adequately conducted and guaranteed in the workplace. In fact, absenteeism due to vaccine preventable diseases is still a current issue, responsible for health and economic losses. This could be solved through two factors: awareness of the problem, and tailored campaigns to prevent it. Leveraging on the first factor, this review offers an international overview, following a rigorous scientific methodology, of vaccination coverage (classified by type of vaccine) in the workplace. The aim is to raise awareness among occupational medicine and public health professionals and to provide a starting point for the resolution of a pervasive problem, and for which there is a compelling scientific need. No similar reviews have been performed and published, to the best of our knowledge.
Reviewer 3 Report
This is an interesting study aiming at evaluating adherence to workplace vaccination campaigns and intending at showing the importance of healthcare workers' involment in the overall campaign success. The implementation of workplace vaccination campaigns, as well as putting in place essential measures to encourage workers to participate in them, would result in a relevant reduction of the risk that exposed workers are subjected to on a daily basis. To my mind, this study may have an impact at the litterature.
Minor Comments
This paper might benefit from english language editing in some parts
Also:
Line 49: Please replace Sars-CoV-2 with SARS-CoV-2 and add abbreviation both for COVID-19 and SARS-CoV-2
Lines 70-71: Please rephrase this sentence “The aim of this research is to perform a systematic review aimed at evaluating the 70 adherence of workers to on-site vaccination campaigns worldwide.”
Author Response
Dear Reviewer,
Thank you for accepting to review our manuscript. Regarding your valuable comments to improve our manuscript, a revision of the English language has been performed thorough the manuscript; the term SARS-CoV-2 has been corrected, extended terms and abbreviations for both OCVID-19 and SARS-CoV-2 have been added (lines 53-54), and the sentence from lines 70-71 has been rephrased.
Round 2
Reviewer 2 Report
Accept in current form